# Rubinstein–Taybi Syndrome Clinical Characteristics from the Perspective of Quality of Life and the Impact of the Disease on Family Functioning

**DOI:** 10.3390/jcm13175210

**Published:** 2024-09-02

**Authors:** Anna Rozensztrauch, Aleksander Basiak, Iwona Twardak

**Affiliations:** 1Division of Paediatrics and Coordinated Child Care, Wroclaw Medical University, 50-367 Wrocław, Poland; iwona.twardak@umw.edu.pl; 2Department of Pediatrics, Endocrinology, Diabetology and Metabolic Diseases, Wroclaw Medical University, 50-367 Wrocław, Poland; aleksander.basiak@umw.edu.pl

**Keywords:** rare disease, Rubinstein–Taybi syndrome, child, family, intellectual disability

## Abstract

**Background/Objectives**: Rubinstein–Taybi Syndrome (RSTS-OMIM, #180849) is a rare genetic disorder associated with distinctive clinical features, including a typical craniofacial appearance, global developmental delay, intellectual disability and broad, angular thumbs and fingers. The main aim of the study was to evaluate the health problems of children with RTST, their quality of life and the impact of the disease on family functioning. In addition, we investigate whether comorbidities, autistic behavior and eating problems affect the children’s overall QOL. **Methods**: A cross-sectional study was performed, including a total of 13 caregivers of children diagnosed with RSTS. A self-reported questionnaire [SRQ], medical records and the Pediatric Impact Module PedsQL^TM^ 2.0, the Pediatric Quality of Life PedsQL^TM^ 4.0 were used to obtain data on QOL and the impact of the disease on family functioning. **Results**: The overall QOL score for children with RSTS was x = 52.40; SD 13.01. The highest QOL was in emotional functioning (EF; x = 59.23; SD 18.69), while the lowest QOL was in physical functioning (PF; x = 48.56; SD 16.32) and social functioning (SF; x = 48.85; SD 18.84). There was a statistically significant negative correlation (*p* < 0.03; r = −2.01) between the age of the child and their QOL, indicating that older children had lower QOL scores. The mean overall rating for the impact of RSTS on family functioning was x = 50.00; SD 10.91. Caregivers reported the highest scores for cognitive functioning (CF; x = 64.23; SD 23.70) and family relationships (FR; x = 60.00; SD 17.17). The lowest scores were for daily activities (DA; x = 41.03; SD 17.17) and worry (W; x = 37.69; SD 18.55). **Conclusions**: This study provides the first comprehensive exploration of the QOL of children with RSTS) and its impact on family functioning.

## 1. Introduction

Rubinstein–Taybi Syndrome (RSTS-OMIM, #180849) is a rare genetic disorder, named after the doctors who first described it, Dr. Jack Rubinstein and Dr. Hooshang Taybi, in 1963 [1]. RSTS is characterized by distinctive clinical features, including a typical craniofacial appearance, global developmental delay, intellectual disability and broad, angular thumbs and fingers [2]. The molecular basis of RSTS was first discovered in the 1990s, when deletions in the region of chromosome 16p13 were first identified in affected individuals, leading to the discovery of pathogenic variants in the *CREBBP* gene [3,4,5,6]. RSTS is divided into two basic types, both autosomal dominant: RSTS type 1 (RSTS1), caused by deletions or pathogenic variants in the *CREBBP* gene and accounts for about 50–60% of RSTS cases [7,8,9] and RSTS type 2 (RSTS2), associated with pathogenic variants in the *EP300* gene, accounting for about 8–10% of RSTS cases [10,11]. Although RSTS2 shares many phenotypic features with RSTS1, there are significant differences. Individuals with RSTS2 tend to have less frequent thumb and toe valgus and experience less severe developmental delays. Approximately 30% of people with RSTS receive a clinical diagnosis based on phenotypic presentation, even though no identifiable pathogenic variant has been found in *CREBBP* or EP300 [12,13]. These genes encode histone acetyltransferases, which are crucial for normal human development through their role in epigenetic regulation [14]. RSTS type 1 is predominantly characterized by three key features: intellectual disability, broad and often oblique thumbs and hallux valgus, and characteristic facial dysmorphia. Characteristic facial features include highly arched eyebrows, downward-sloping eyelid crevices, a wide nasal ridge, a column that hangs below the alae nasi, a thin upper lip, a puffed-out lower lip and mild micrognathia (a small lower jaw). In addition to these characteristic features, several other variable features have been reported in people with RSTS type 1. These may include heart defects, a highly arched palate (with or without a cleft uvula), low-set ears that are rotated backward, dislocation of the patella, reduced immune function and undescended testes in males. Moreover, patients with RSTS type 1 have an increased risk of developing cancers, especially those affecting the nervous system [10,12,15].

Children with RSTS often experience a range of behavioral challenges that can significantly affect their quality of life (QOL). Early studies characterized these children as typically having a friendly and excitable temperament that is commonly associated with hyperactivity, emotional dysregulation, short attention spans, self-stimulatory behaviors, and difficulties in planning and performing motor tasks. Stevens et al. [9] and Wiley et al. [16] found that obsessive compulsive disorder, characterized by recurrent and disturbing obsessions and repetitive behaviors, is often a concurrent diagnosis with RSTS and can impair an individual’s ability to interact with others and perform the daily life activities. The studies by Verhoeven et al. [17] and Waite et al. [18] confirmed some previously known behavioral features in children with RSTS, such as hyperactivity, emotional dysregulation and self-stimulatory behavior. In addition, their findings introduced a further significant psychological phenotype: anxiety. This suggests that anxiety is a significant and potentially underexplored aspect of the behavioral profile in children with RSTS, further contributing to the ongoing challenges they face. Crawford et al. [19] found that children with RSTS had similar levels of anxiety on the panic attack and agoraphobia subscales and obsessive compulsive subscales of the Spence Child Anxiety Scale—Parent Version compared to children diagnosed with anxiety disorder. 

Understanding the QOL of children with RSTS is essential to developing effective supportive interventions that can improve their well-being and help them lead satisfying lives [20]. QOL is focused on the stage to which children experience quality of life in their family environment, as well as how the family has the potential to achieve its goals in the community and society. RSTS significantly affects the QOL of affected children and their families, as emphasized by much research [9,21,22]. Children with RSTS often demonstrate a range of physical and intellectual disabilities, these delays require constant medical care, specialized educational programs and customized therapeutic interventions, which can be physically, emotionally and financially demanding for families [23,24]. Research on QOL and the impact of disease on family functioning focuses on the main areas of life activities that are important to family functioning and which are usually referred to as dimensions or domains. Taken together, these various domains, such as family interaction, parenting, emotional well-being, physical/material well-being and peer support, represent family life in general [25]. Children with RSTS frequently undergo a wide range of health problems, such as heart defects, breathing problems and feeding difficulties. Regular medical consultations, hospitalizations and surgeries are common, increasing the family’s stress and emotional burden. Mori et al. [26] examined family QOL among families with a child with a CDKL5 disorder and found that it was generally lower among those receiving social support, such as foster care services, suggesting that these families may be more burdened by daily care.

The main aim of the study was to evaluate the health problems of children with RSTS, their quality of life and the impact of the disease on family functioning. In addition, we investigate whether the comorbidities, autistic behavior and eating problems affect the children’s overall QOL. The final aim was also to determine the impact of frequent hospitalizations and parents’ inability to rest on the QOL and functioning of the whole family.

## 2. Materials and Methods

### 2.1. Setting

A cross-sectional study was performed, including a total of 13 caregivers of children diagnosed with Rubinstein–Taybi Syndrome (RSTS). The children included in the study were patients of the Genetic Outpatient Clinic of the Foundation for Children with Rare Disease Disorders. The study was conducted during a follow-up visit. All participants received a complete set of questionnaires to fill out, and each provided written informed consent to participate in the study. The parents were included in the study if they met the following inclusion criteria: they are the biological parents of the child; their child was diagnosed with RSTS by a pediatrician and/or neurologist, with the diagnosis confirmed clinically by molecular testing.

The study was conducted according to the guidelines of the Declaration of Helsinki and approved by the Institutional Review Board (or Ethics Committee) of Wroclaw Medical University (protocol code KB 35/2021 and 29 January 2021).

### 2.2. Study Sample

The average age of the children involved in the study was 7.9 (SD = 4.1). The youngest child was 3 years old, while the oldest was 17 years old. Most children (39%) were in the age range of 5–7 years, 54% of the children studied were boys, while 46% were girls.

The study involved 13 parents, the mothers averaged 40.2 ± 9.5 years (minimum age 23 years and maximum age 65 years) and the fathers 42.1 ± 9.7 years (minimum age 26 years and maximum age 71 years); only one caregiver was a single parent.

### 2.3. Clinical Characteristics of the Prenatal and Perinatal Period of the Study Children

The study children were diagnosed with RSTS in the 1st year of life (40%), the 2nd year of life (15%), the 3rd year of life (15%), the 4th year of life (15%) or the 5th year of life (8%). The majority (38%) of the children were born at 36 gestation weeks. Almost half (46%) of the parents first noticed RSTS symptoms right after birth. In 62% of children, complications developed during or after delivery. The most common of these were: hypoxia (37.5%), jaundice (37.5%), pneumonia (50%) and respiratory distress syndrome (25%). The defects diagnosed right after birth included: laryngomalacia (18%), cryptorchidism (9%), respiratory failure (9%), multiple organ failure (9%), excessive hair growth (9%), vascular ring (9%), mandibular hypoplasia (9%), high-arched palate (27%), Chiari malformation (9%), esotropia (9%), astigmatism (9%), hearing impairment (9%), craniosynostosis (9%), microcephaly (9%), phimosis (9%), pes calcaneus (9%), syndactyly (9%), bicuspid aortic valve (9%), auricular malformations (9%) and facial dysmorphic features (45%). A total of 54% of patients had to undergo surgery due to the defects. Calculated BMI in five children (38.5%) were underweight (3 girls aged 3, 7 and 9 and 2 boys aged 5 and 4). In total, 2 children were overweight (15%) (2 boys aged 4 and 7). One child (8%) was obese (a boy aged 11). A total of 32% of patients had weight deficiency. Congenital heart defects were reported in three patients (23%). These included the following: type 2 atrial septal defect, false tendon, and bicuspid aortic valve. A total of 23% of caregivers reported that their children suffered from recurrent urinary tract infections. Kidney defects were reported in one child. None of the boys included in the study had hypospadias. Two caregivers reported that their children with obstructive sleep apnoea are tired, irritable and hyperactive during the day. In a total of 38% of the children included in the study, sleep problems other than obstructive sleep apnoea were reported. Four parents cited sudden and frequent wakeups with difficult-to-soothe crying as the main sleep problem experienced by their children. Patient characteristics are shown in Table 1.

### 2.4. Instruments

A self-reported questionnaire [SRQ] and medical records were used to collect socio-demographic data and clinical features (comorbidities, genes, motor development). Standardized questionnaires: the Pediatric Impact Module PedsQL^TM^ 2.0 and the Pediatric Quality of Life PedsQL^TM^ 4.0 were used to obtain data on QOL and the impact of the disease on family functioning.

### 2.5. The Pediatric Impact Module PedsQL^TM^ 2.0–PedsQL^TM^–FIM

The PedsQL™-FIM measures family functioning and is designed to measure the impact of chronic pediatric conditions on parents and families. The instrument is composed of 36 items measuring parents’ self-reported family functioning in six domains: physical functioning (PF-6 items), emotional functioning (EF-5 items), social functioning (SF-4 items), cognitive functioning (CF-5 items), communication (C-3 items) and worry (W-5 items); two additional domains measure parents’ reported family functioning: daily activities (DA-3 items) and family relationships (FR-5 items). Each item is rated using a 5-point Likert scale from 0 (never a problem) to 4 (always a problem), which is then converted into a scale from 0 to 100 (0 = 100, 1 = 75, 2 = 50, 3 = 25, 4 = 0), with higher scores representing better functioning [18].

### 2.6. The Pediatric Quality of Life PedsQL^TM^ 4.0

The Paediatric Quality of Life Inventory (PedsQL) is a validated instrument to measure QOL in children and adolescents aged 2–18 years with acute or chronic medical conditions. The instrument contains a questionnaire to measure four dimensions of functional outcomes: physical functioning (PF), emotional functioning (EF), social functioning (SF) and school functioning (SCHF). A five-point response scale is used (0 = never a problem, 4 = almost always a problem). A higher score on the PedsQL questionnaire points to a higher QOL. The instrument has good psychometric properties (Cronbach’s alpha coefficient ranges from 0.66 to 0.93) [27,28,29,30].

### 2.7. Statistical Analysis

Analysis of quantitative variables—those that can be expressed numerically—was conducted by calculating the mean, standard deviation, median, quartiles, minimum and maximum. Analysis of qualitative variables—those that cannot be expressed numerically—was conducted by calculating the number and percentage of occurrence of each value. Comparisons of the values of qualitative variables were made using Fisher’s exact test. Comparisons of the values of quantitative variables in the two groups were made using the Student’s t-test when the variable had a normal distribution or the Mann–Whitney–Wilcoxon U test when it did not. Comparisons between quantitative variables in the three groups were made by analysis of variance (ANOVA). Correlation analysis between quantitative variables was performed using Pearson’s correlation coefficient method when the variables had a normal distribution or Spearman’s correlation coefficient when the variables did not have a normal distribution. The normality of the distribution of quantitative variables was tested using the Shapiro–Wilk test, while the homogeneity of variance was checked using the Levene test. A significance level of 0.05 was assumed for the analysis, so all *p*-values below this value were considered to indicate the presence of significant relationships. Calculations were performed using R software version 3.6.1.

## 3. Results

### 3.1. QOL of Children with Rsts

The overall quality of life was at $\bar{x}$ = 52.40 (standard deviation, SD 13.01). The children experienced the highest QOL in emotional functioning (EF) (EF; $\bar{x}$ = 59.23, SD 18.69) and the lowest QOL in physical functioning (PF) (PF $\bar{x}$ = 48.56, SD 16.32) and social functioning (SF) (SF $\bar{x}$ = 48.85, SD 18.84) (Table 2). 

We found a statistically significant correlation (*p* < 0.03; −2.01) between the age of the child and their QOL (*p* < 0.03; r = −2.01), the older the child, the lower their QOL (Table 3).

Irritability, stubbornness, lack of persistence and sudden changes in mood are behavioral impairments that occur in 92% of the children participating in the study. No statistically significant correlation was found between the presence of comorbidities and overall QOL (*p* > 0.16, RHO −0.29), sleep apnoea significantly reducing QOL of the child (*p* < 0.001) All the children included in the present study were aged over 1 year. However, 6 (46%) of them still had eating problems. These included the following: problems with chewing (56%), food selectivity (22%), loss of appetite (11%) and problems with swallowing (11%). Moreover, 7 children (54%) experienced choking episodes and 6 children (46%) experienced vomiting. Heartburn was reported in only 3 (23%) of the patients studied. Our study revealed a correlation (*p* < 0.001) between the presence of vomiting and choking and reduction in QOL (PedsQOL *p* = 0.001582, PedsQOL–FIM, *p* = 0.1764) (Table 4). Furthermore, there was no correlation between the presence of heartburn and a reduction in the QOL of the children and their families (*p* > 0.05). 

### 3.2. The Family Impact—Pedsql^tm^–Fim

The mean overall rating for the impact of RSTS on family functioning was $\bar{x}$ = 50.00 (SD = 10.91). The scores reported by caregivers were highest for cognitive functioning (CF, $\bar{x}$ = 64.23, SD = 23.70), family relationships (FR, $\bar{x}$ = 60.00, SD = 17.17) and lowest for the daily activities (DA, $\bar{x}$ = 41.03, SD = 17.17) and worry (W, $\bar{x}$ = 37.69, SD = 18.55) domains (Table 5).

No significant correlation (*p* > 0.05) was found between a larger family size and a reduction in overall QOL. However, the opposite was found for the QOL of the family. The larger the family size, the higher the QOL. However, the relationship was not statistically significant (*p* > 0.05) (Table 6). We found no correlation between the age of the child and the QOL of their family (*p* > 0.22; r = −0.76).

We found a statistically significant correlation between the presence of autistic-like behaviors and a reduction in the overall QOL of the children (*p* = 0.00979) and their families (*p* = 0.02517). We found a correlation (*p* < 0.02) between support from family and friends in caring for a child with RSTS and higher QOL. No relationship was demonstrated between a deterioration in family relations and a reduction in QOL (*p* > 0.05).

## 4. Discussion

While studies on the QOL of patients with common diseases are readily available, studies focusing on the QOL of patients with rare diseases are much more difficult to find. This study included a highly diverse group of RSTS patients and their parents, emphasizing the unique challenges and variability in this group. Children differed in age, comorbidities, degree of disability, age at diagnosis and living environment. Similarly, parents varied in age, education, environment, access to specialists, and, most critically, attitudes toward their child’s illness and level of acceptability. 

The study found that the overall QOL for children with Rubinstein–Taybi Syndrome is markedly low. While these children generally perform best in the emotional domain, they experience the most significant challenges in the physical domain. A study reported by Miller et al. [31] assessed the QOL of children with developmental disorders, showing a growing tendency in which emotional aspects are relatively better, while physical health remains a challenge. In a study carried out by Bidzan et al. [21], the child studied had high scores on the emotionality subscale, medium scores on the activity subscale, medium scores on the sociability subscale and low scores on the shyness subscale of the EAS Temperament questionnaire. An analysis of the scores reported by the parents on the Satisfaction with Life Scale questionnaire indicated that the parents rated their QOL as average. The children with RSTS included in the present study experienced the highest QOL in emotional functioning. The scores reported for the children by their parents were highest for emotional functioning and lowest for physical functioning. The scores reported by the parents on the family questionnaire were highest for cognitive functioning and family relationships and lowest for the worry and daily activities domains. Our study found that the older the child with RSTS, the lower their QOL score. According to Von der Lippe et al. [32], age is one of the most important determinants of QOL. As children with RSTS grow older, the cumulative effects of intellectual disability, physical limitations, and social challenges may become more pronounced, leading to a progressive decline in QOL. This age-related trend underscores the need for continuous support and tailored interventions that evolve with the child’s age and changing needs. In a study by Agborsangaya et al. [33], associations between multimorbidity and health-related quality of life were assessed. In that study, individuals with at least one chronic condition were found to have significantly lower QOL compared with individuals without chronic conditions. Moreover, individuals with multiple (two or more) chronic conditions had significantly low scores. The study also found that individuals with multimorbidity were twice as likely to be hospitalized compared with the healthy individuals included in the study. In the present study, we hypothesised that the presence of a higher number of medical disorders and more frequent hospital stays in children with RSTS can affect the QOL of the children and their families. The study did not confirm the hypothesis. However, we found that children with eating difficulties (vomiting, choking) have a significantly lower QOL. 

Children with RSTS have dysmorphic facial, hand and foot features. They also have intellectual and psychomotor delays. In some children, speech never develops at all and some children have behavioral problems [34]. Studies have found that approximately 25% of parents of children with RSTS report behavioral problems in their children. These mainly include short attention span, stubbornness, lack of persistence, continuous need for attention, sudden mood changes, resistance to change and repetitive movements [35]. In Grunow et al.’s [36] study, as many as 65% of children with RSTS were reported to display such unusual behaviors as rocking, spinning and hand flapping. Some children with RSTS exhibit autistic-like behaviors. In the present study, behavioral problems (e.g., irritability, stubbornness, lack of persistence, sudden mood changes) were reported in the children studied. Autism-like behaviors can increase the already severe symptoms of RSTS, potentially leading to more severe difficulties in social interactions, communication and daily functioning. These behaviors likely contribute to added stress and emotional strain on the family, further compromising their overall QOL. The positive correlation between receiving support from family and friends and higher QOL might be a positive contribution. This finding is consistent with the existing work on chronic disease, which consistently shows that social support has a critical role in alleviating stress and improving overall well-being. For families of children with RSTS, support from extended family, friends and community resources can provide much-needed respite, emotional encouragement and practical help, thereby enhancing their ability to cope with daily challenges. However, the lack of a demonstrated correlation between worsening family relationships and reduced QOL is rather intriguing (*p* > 0.05), in contrast to our expectations. This suggests that while the presence of a child with RSTS clearly affects family dynamics, the direct impact on QOL may be mitigated by other variables, such as dealing strategies, external support and individual family immunity. This finding highlights the need for further research into the specific aspects of family relationships that are most impacted and how they interact with other support systems to influence QOL. Together, these insights underscore the multi-dimensional nature of living with RSTS and the importance of a holistic approach to support. Healthcare providers should be aware of the significant impact of autism-like behaviors and prioritize interventions that address these symptoms. Additionally, qualitative study could reveal the nuanced ways in which family dynamics interact with other factors affecting QOL, offering a more comprehensive understanding of the lived experiences of these families. Children with RSTS normally experience intellectual delay, and studies [37] show that the average IQ of patients with RSTS ranges from 35 to 50. In our study, we determined the degree of intellectual disability plays a significant role in the overall QOL of children. Intellectual delay can affect various aspects of daily life, including education, social interaction and independence, thus contributing to the lower QOL observed in our participants. The physical symptoms of RSTS play a role in QOL. In a series of studies of people with RSTS published in 1990 by Rubinstein [1], wide thumbs and/or halluces were reported in 100% of cases. Other commonly found findings included angular deformity of the thumbs with abnormal proximal phalanx shape, angular deformity of the halluces with abnormal shape of the proximal phalanx or first metatarsal bone, clinodactyly of the fifth toe, overlapping toes and broad terminal phalanges in other toes. These physical abnormalities can contribute to functional limitations and social stigmatization, further affecting the QOL for the child and his family.

Since children with RSTS suffer from several congenital anomalies and medical disorders, their caregivers must regularly monitor the health status of their children and have regular contact with many specialists in different medical fields. The medical disorders affecting children with RSTS result in 10 times the average number of hospital stays as the general population [38]. Considering the poor weight gain often observed in children with RSTS, it is essential to monitor their growth from birth through regular measurement of weight, height and head circumference. In addition, young patients with RSTS need to be subjected to a variety of specialized tests to meet their specific health needs. Due to the psychomotor delays associated with RSTS, it is often necessary to integrate rehabilitation into the treatment process [39]. 

The present study showed that the families of children exhibiting autistic-like behaviors, such as failure to respond to instructions, lack of peer play and difficulties expressing emotions. The same relationship was found for the child’s QOL. Children with autistic-like behaviors have lower QOL. In a study by Czenczek et al. [40], the QOL of parents of children with autism was examined. In that study, parents of autistic children reported poorer functioning in the areas of general health, mental health, social functioning, energy and vitality. A study by Lee et al. [41] also found that autism has a negative impact on the quality of life of the families of affected children. The study found that the burden of care reported by families with autistic children was significantly higher relative to the two comparison groups. 

Support from family and friends significantly improves the QOL of both children with RSTS and their parents. Parents who received such support reported higher family impact scores. Marie et al. [31] emphasized that close family members are key to successful treatment and rehabilitation, highlighting the importance of a strong social support network. Similarly, Jamieson et al. [42] found that families with children with disabilities often experience low QOL, particularly in terms of social support. Our results are consistent with these studies, showing that well-established support can ease stress and improve the overall well-being of families of children with RSTS. As many as 51% of the patients included in the study reported experiencing an increased level of sleepiness during the day, which significantly impaired their functioning. The authors noted that uninterrupted sleep is essential for normal functioning. In the present study, we hypothesised that obstructive sleep apnoea in children with RSTS has a negative impact on the QOL of the children and the functioning of their families. 

Research on QOL, especially among all patient groups, not just those with common conditions, is essential. Such studies help identify factors that affect QOL and enable the development of targeted measures to improve it. Understanding the diverse experiences and needs of families affected by RSTS can provide information on better support, health care policies and interventions, ultimately improving their quality of life. Moreover, it should be noted that parents of children with disabilities often feel tired and burdened from constantly caring for their children. Many parents do not receive support from their family and friends and are left on their own, which significantly constrains the time they have to themselves. 

### 4.1. Limitations

The study group was not large. However, it should be noted that RSTS is a rare genetic disorder, occurring in about 1 in 100,000 to 125,000 births. Considering the prevalence of RSTS, this represents a significant sample size. Moreover, it is important to consider the serious challenges faced by parents of children with disabilities, including those with RSTS. These parents often experience significant fatigue and strain due to the constant care their children require. Many parents do not receive sufficient support from family and friends, causing them to have to handle caregiving responsibilities on their own. This deficiency in support significantly limits their personal time, which makes their participation in our study even more remarkable.

### 4.2. Therapeutic Recommendations for Children with Rubinstein–Taybi Syndrome

Systematic reassessment of the child’s development and the effectiveness of interventions is necessary, allowing the therapeutic plan to be adjusted as the child grows and his needs change. Therapy should be initiated as early as possible to support cognitive, motor, and social development. Early intervention is crucial for improving long-term outcomes and maximizing the child’s potential. A multidisciplinary care approach is essential, involving a team of specialists such as pediatricians, geneticists, neurologists, speech therapists, occupational therapists, nursing care and physical therapists. Physical therapy focuses on strengthening muscles, improving motor skills, and enhancing overall mobility. Occupational therapy aims to enhance fine motor skills, daily living activities, and sensory processing. It involves targeted activities to improve hand-eye coordination, feeding abilities, and self-care skills. Sensory integration therapy can also be effective in managing sensory processing challenges. Speech and language therapy is essential for advancing both verbal and non-verbal communication skills and addressing issues related to feeding and swallowing. 

## 5. Conclusions

This study provides the first comprehensive exploration of the QOL of children with RSTS and its impact on family functioning. Our findings highlight the significant and ongoing health challenges faced by children with RSTS, encompassing physical, emotional, and social difficulties, which collectively contribute to a lower overall QOL. Furthermore, the study reveals the profound impact of the syndrome on family dynamics, with heightened emotional strain among family members. These insights underscore the urgent need for targeted interventions and support systems to enhance the QOL for both children with RSTS and their families. Future research should prioritize the development and evaluation of such interventions, as well as the examination of long-term outcomes as these children transition into adulthood.

## Figures and Tables

**Table 1 jcm-13-05210-t001:** Study sample characteristics.

	1	2	3	4	5	6	7	8	9	10	11	12	13
Gene	CREBBP	CREBBP	CREBBP	CREBBP	EP300	EP300	EP300	EP300	CREBBP	CREBBP	CREBBP	CREBBP	CREBBP
Diagnosis	18th day	3 years	1 year	2 years	4 years	1 year	2 months	5 years	1 year	2 years	5 years	4 years	2.5 years
Current age	7 years	5 years	11 years	3 years	9 years	7 years	11 years	17 years	4 years	7 years	5 years	4 years	13 years
Child’s weight	25–50th centile	3rd centile	50–75th centile	below the 3rd centile	10th centile	below the 3rd centile	90–97th centile	below the 3rd centile	50–70th centile	50–75th centile	25–50th centile	10–25th centile	below the 3rd centile
Child’s height	3rd–10th centile	3rd–10th centile	25th centile	below the 3rd centile	10–25th centile	below the 3rd centile	3rd–10th centile	below the 3rd centile	10–25th centile	below the 3rd centile	3rd centile	90th centile	below the 3rd centile
BMI	16.9 normal	13.6 underweight	19.8 normal	13.2 underweight	14.6 underweight	13.3 underweight	31.4 obesity	18.4 normal	17.1 overweight	19.6 overweight	17 normal	12.4 underweight	17.8 normal
Child’s head circumference	below the 3rd centile	below the 3rd centile	below the 3rd centile	10th centile	below the 3rd centile	above the 99th centile	-	23rd centile	25–50th centile	7th centile	25–50th centile	25–50th centile	62nd centile
Prenatal course	NAD	NAD	NAD	abnormal foetal size	NAD	NAD	NAD	NAD	NAD	abnormal amniotic fluid volume, pregnancy at risk of miscarriage	pregnancy at risk of miscarriage	NAD	NAD
HBD	39 weeks	36 weeks	40 weeks	36 weeks	36 weeks	35 weeks	39 weeks	39 weeks	40 weeks	38 weeks	39 weeks	36 weeks	36 weeks
Birth weight (g)	10–50th centile	50th centile	10th centile	10–50th centile	50th centile	50th centile	below the 3rd centile	3rd centile	10th centile	10–50th centile	below the 3rd centile	50th centile	50th centile
Postpartum period	jaundice, pneumonia, feeding problems	hypoxia, respiratory distress syndrome, pneumonia, feeding problems	hypoxia	feeding problems	jaundice, feeding problems	respiratory distress syndrome, vascular ring, feeding problems	feeding problems, reflux	hypoxia, pneumonia, one lung was not functioning, feeding problems	genital defects, feeding problems	jaundice, feeding problems	feeding problems	second superior vena cava, feeding problems	pneumonia, feeding problems
Motor development	delayed	delayed	delayed	delayed	delayed	delayed	delayed	delayed	delayed	delayed	delayed	delayed	delayed
Speech development	normal	lack of speech development	lack of speech development	-	normal	delayed	delayed	normal	lack of speech development	normal	normal	delayed	delayed
Intellectual disability	moderate	moderate	moderate	moderate	moderate	moderate	significant	moderate	moderate	moderate	mild	mild	moderate
Typical facial dysmorphic features	+	+	+	+	+	+	+	+	+	+	+	+	+
Typical dysmorphic limb features–hands, fingers (including thumbs) and feet	+	+	+	+	+	+	+	+	+	+	+	-	+
Typical behavioral phenotype	+	+	-	+	+	+	+	+	+	+	+	+	+
Hearing	bilateral impairment	bilateral impairment	-	-	-	-	-	-	-	-	-	bilateral hearing impairment	-
Eyesight	astigmatism, myopia	astigmatism, esotropia	hyperopia	strabismus, myopia	-	astigmatism, hyperopia	-	astigmatism, hyperopia	esotropia, astigmatism	hyperopia	-	optic nerve hypoplasia	astigmatism, esotropia, hyperopia
Keloid formation	+	+	-	-	-	-	+	-	-	-	-	+	-
Genital abnormalities	-	+	-	+	-	-	+	-	+	+	-	-	-
Kidney defects	+	-	-	-	-	-	-	-	-	-	-	-	-
Heart defects	+	+	-	-	-	-	-	-	-	-	-	+	-
Short stature	-	-	+	+	-	+	+	-	+	+	+	-	+
Abnormalities of hands, including thumbs	+	+	-	+	-	+	+	+	-	+	+	-	+
Abnormalities of feet, including halluces	+	+	-	-	-	+	+	+	-	+	+	-	-
Occlusal abnormalities/dental anomalies	-	+	+	+	+	+	-	+	+	+	+	+	-
Brain MRI scan	hypoplasia of the P1 segment of the left posterior cerebral artery	NAD	NAD	NAD	small non-specific subcortical hyperintense focus in the posterior right parietal lobe	NAD	NAD	NAD	NAD	NAD	delayed white matter myelination	underdevelopment of optic nerves and the optic chiasm, underdevelopment of the corpus callosum	hypoplasia of the cerebellar vermis, cysts

**Table 2 jcm-13-05210-t002:** Scores in the PedsQL^TM^-4.0.

PedsQL	N	Mean	SD	Median	Min.	Max.	Q1	Q3
PF	13	48.56	16.32	50.00	18.75	78.13	37.50	56.25
EF	13	59.23	18.69	60.00	25.00	100.00	50.00	70.00
SF	13	48.85	18.84	45.00	20.00	85.00	40.00	60.00
SCHF	12	50.74	17.68	50.00	20.00	85.00	40.15	61.25
PSF	13	53.31	14.44	53.33	28.33	78.33	46.67	61.67
Total QoL score	13	52.40	13.01	51.41	29.84	71.25	45.24	61.88

$\bar{x}$ mean; Me, median; Max, maximum value; Min, minimum value; N, number of respondents; Q1, first quartile; Q3, third quartile; SD, standard deviation; PF, physical functioning; EF, emotional functioning; SF, social functioning; SCHF, school functioning PSF, Psychosocial functioning.

**Table 3 jcm-13-05210-t003:** Age correlation scores in the PedsQL^TM^-4.0, including Pearson’s coefficient.

PedsQL	Pearson’s Coefficient	*p* *
Child’s total QoL score	−2.0185	0.0343
Total impact score (PedsQL Family Impact Module)	−0.76871	0.2291

* Statistically significant relationship (*p* < 0.05).

**Table 4 jcm-13-05210-t004:** PedsQOL scores and presence of vomiting and choking.

PedsQL	*p* *
Child’s total QoL score	0.001582
Total impact score (PedsQL Family Impact Module)	0.1764

* Statistically significant relationship (*p* < 0.05).

**Table 5 jcm-13-05210-t005:** Impact of RSTS on family functioning—PedsQL–FIM.

PedsQL-Family Impact								
Module	n	Mean	SD	Median	Min.	Max.	Q1	Q3
PF	13	44.55	15.44	45.83	16.67	70.83	33.33	54.17
EF	13	46.54	14.05	45.00	20.00	70.00	40.00	55.00
SF	13	52.40	20.50	56.25	18.75	93.75	43.75	68.75
CF	13	64.23	23.70	65.00	25.00	100.00	45.00	75.00
C	13	52.56	14.98	58.33	25.00	75.00	41.67	58.33
W	13	37.69	18.55	30.00	10.00	65.00	25.00	50.00
DA	13	41.03	17.17	41.67	16.67	75.00	25.00	50.00
FR	13	60.00	12.91	60.00	40.00	85.00	55.00	65.00
Parent QoL	13	51.54	13.46	50.00	30.00	75.00	42.50	65.00
FF	13	52.88	11.93	56.25	34.38	75.00	43.75	59.38
Total impact score	13	50.00	10.91	51.39	31.25	68.75	42.36	56.25

$\bar{x}$

, mean; Me, median; Max, maximum value; Min, minimum value; N, number of respondents; Q1, first quartile; Q3, third quartile; SD, standard deviation; PF, physical functioning; EF, emotional functioning; SF, social functioning; CF, Cognitive functioning; C, Communication; W, Worries; DA, Daily activities; FR, Family relationships; FF, Family functioning.

**Table 6 jcm-13-05210-t006:** Family size and PedsQOL, including Spearmen’s rho coefficient.

PedsQL	r	*p* *
Child’s total QoL score	−0.2964997	0.1626
Total impact score (PedsQL FIM)	0.4076871	0.08336

* Statistically significant relationship *p* < 0.05.

## Data Availability

The data that support the findings of this study are available from the corresponding author, upon reasonable request.

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
