# Peer review of "Rubinstein–Taybi Syndrome Clinical Characteristics from the Perspective of Quality of Life and the Impact of the Disease on Family Functioning"

_jcm, 2024, doi:10.3390/jcm13175210_

Round 1

Reviewer 1 Report

Comments and Suggestions for Authors

[JCM] Manuscript ID: jcm-3126328 - Review Report

I would like to thank you for the opportunity to review this paper, which delves into a topic dear to me: quality of life and the impact of illness on family functioning in rare and complex diseases.The care of a child with disability must necessarily involve the whole family in the treatment plan, in line with the theoretical assumptions of the Family Centered Care (FCC) model. where the family is co-partner in treatment."  

Inizio modulo

Neverthless significant revision are needed, both in form and conceptual issues. Considering this reasons I have significant concerns about its potential impact due. My perception is that the methodology is confused and some of the results are already well known. The sample is really small despite the rarity of the condition nd this impacts on the significance of the data. The presentation of the manuscript could benefit of a considerable revision too.

Here below some more specific comments for the author to consider :

INTRODUCTION

The introduction could be enhanced with revisions and reorganization. I find it confusing, and the organization can be improved to better explain the focus of the paper. It is necessary to rewrite and reorganize the introduction, taking into account a comprehensive review of the literature to justify these correlations. Here are some specific points:

  1. The description of the syndrome needs to be revised by updating the literature data. It is now well-known that there are two pathogenic variants: CREBBP leads to RSTS type 1 (RSTS1), and EP300 leads to RSTS type 2 (RSTS2). These variants have specific characteristics in terms of clinical and behavioral phenotypes, which should be described. These characteristics can impact the well-being and quality of life of affected children and their families. Is it also necessary to describe the prevalence of these two conditions

  2. It is necessary to systematize and better define the purpose of the work. Decisions need to be made regarding which clinical and behavioral elements to focus on in order to assess their impact on wellbeing. For example, individuals with RSTS can demonstrate a variety of behavioral and neuropsychiatric challenges, including anxiety, hyperactivity/inattention, self-injury, repetitive behaviors, and aggression. Behavioral challenges consistently impact quality of life.

Each chosen variable (e.g., presence of malformations, hospitalizations, BMI, behavioral phenotype, psychosocial family situation, age) should be justified with recent literature.

On page 2, line 54, the sentence “RSTS significantly affects the QoL… as emphasized by much research” needs to be supported by references  

METHODS

The “Methods” section also needs to be revised and organized. I suggest to create a paragraph titled “The Sample,” where both children and parents are better described.

For example for the children:

  • Age (mean and standard deviation)

  • Gender

  • Genes (EP300 or CREBBP)

  • Intellectual disability (yes or no)

  • Autism spectrum disorder (yes or no)

  • Presence of malformations (yes or no)

  • …..

We currently lack information about these children, including their age range.

Perhaps creating a table would be helpful.

For the parents:

  • Age (for instance, literature on chronic diseases suggests that older parents experience more anxiety and stress than younger parents)

  • Are responses to the questionnaire collected only from mothers or also from fathers?

  • Single-parent families?

  • Educational background and socio-cultural level

  • …..

RESULTS

The entire “Clinical Characteristics” section ( PAG.3 , line137) should be moved to the “Methods” section.

DISCUSSION

The discussion could benefit from revisions and reorganisations. It should be improved to better explain and discuss the document’s findings. I recommend following the results model, discussing in depth the different questionnaires and correlations, including strengths and weaknesses.

In addition, management and therapeutic recommendations should be provided.

ABSTRACT

The abstract will have to be rewritten after substantial changes to the structure of the article.

ADDITIONAL ELEMENTS

Examine the bibliography carefully.

For example: several references need to be corrected, such as the numbers 24 and 25.

Comments on the Quality of English Language

Minor editing of English language required

Author Response

Dear Reviewer 1,

Thank you very much for sending us the consensus opinion about requested revision of our manuscript entitled: Rubinstein-Taybi syndrome clinical characteristics from the perspective of quality of life and the impact of the disease on family functioning. We appreciate the thoughtful comments, and we have modified the manuscript in response to your suggestions, which we believe will further improve its quality. 

REVIEWER COMMENTS 1

I would like to thank you for the opportunity to review this paper, which delves into a topic dear to me: quality of life and the impact of illness on family functioning in rare and complex diseases. The care of a child with disability must necessarily involve the whole family in the treatment plan, in line with the theoretical assumptions of the Family Centered Care (FCC) model. where the family is co-partner in treatment."  Neverthless significant revision are needed, both in form and conceptual issues. Considering this reasons I have significant concerns about its potential impact due. My perception is that the methodology is confused and some of the results are already well known. The sample is really small despite the rarity of the condition nd this impacts on the significance of the data. The presentation of the manuscript could benefit of a considerable revision too.

Thank you very much for this comment.

REVIEWER COMMENTS 2

INTRODUCTION

The introduction could be enhanced with revisions and reorganization. I find it confusing, and the organization can be improved to better explain the focus of the paper. It is necessary to rewrite and reorganize the introduction, taking into account a comprehensive review of the literature to justify these correlations. Here are some specific points:

  1. The description of the syndrome needs to be revised by updating the literature data. It is now well-known that there are two pathogenic variants: CREBBP leads to RSTS type 1 (RSTS1), and EP300 leads to RSTS type 2 (RSTS2). These variants have specific characteristics in terms of clinical and behavioral phenotypes, which should be described. These characteristics can impact the well-being and quality of life of affected children and their families. Is it also necessary to describe the prevalence of these two conditions
  2. It is necessary to systematize and better define the purpose of the work. Decisions need to be made regarding which clinical and behavioral elements to focus on to assess their impact on wellbeing. For example, individuals with RSTS can demonstrate a variety of behavioral and neuropsychiatric challenges, including anxiety, hyperactivity/inattention, self-injury, repetitive behaviors, and aggression. Behavioral challenges consistently impact quality of life.

 Each chosen variable (e.g., presence of malformations, hospitalizations, BMI, behavioral phenotype, psychosocial family situation, age) should be justified with recent literature.

 Thank you for this comment. Please see the following rewritten introduction section.

“…Rubinstein-Taybi Syndrome (RSTS- OMIM, #180849) is a rare genetic disorder named after the doctors who first described it, Dr. Jack Rubinstein and Dr. Hooshang Taybi, in 1963 [1]. RSTS is characterized by distinctive clinical features, including a typical craniofacial appearance, global developmental delay, intellectual disability and broad, angular thumbs and fingers [2]. The molecular basis of RSTS was first discovered in the 1990s, when deletions in the region of chromosome 16p13 were first identified in affected individuals, leading to the discovery of pathogenic variants in the CREBBP gene [3-6]. RSTS is divided into two basic types, both autosomal dominant: RSTS type 1 (RSTS1), caused by deletions or pathogenic variants in the CREBBP gene and accounts for about 50%-60% of RSTS cases [7-9] and RSTS type 2 (RSTS2, associated with pathogenic variants in the EP300 gene, accounting for about 8%-10% of RSTS cases [10 -11]. Although RSTS2 shares many phenotypic features with RSTS1, there are significant differences. Individuals with RSTS2 tend to have less frequent thumb and toe valgus and experience less severe developmental delays. Approximately 30% of people with RSTS receive a clinical diagnosis based on phenotypic presentation, even though no identifiable pathogenic variant has been found in CREBBP or EP300 [12, 13]. These genes encode histone acetyltransferases, which are crucial for normal human development through their role in epigenetic regulation [14]. RSTS type 1 is predominantly characterized by three key features: intellectual disability, broad and often oblique thumbs and hallux valgus, and characteristic facial dysmorphia. Characteristic facial features include highly arched eyebrows, downward sloping eyelid crevices, a wide nasal ridge, a column that hangs below the alae nasi, a thin upper lip, a puffed-out lower lip and mild micrognathia (a small lower jaw). In addition to these characteristic features, several other variable features have been reported in people with RSTS type 1. These may include heart defects, a highly arched palate (with or without a cleft uvula), low-set ears that are rotated backward, dislocation of the patella, reduced immune function and undescended testes in males. Moreover, patients with RSTS type 1 have an increased risk of developing cancers, especially those affecting the nervous system [10,12,15].

Children with RSTS often experience a range of behavioral challenges that can significantly affect their quality of life (QOL). Early studies characterized these children as typically having a friendly and excitable temperament that is commonly associated with hyperactivity, emotional dysregulation, short attention spans, self-stimulatory behaviors, and difficulties in planning and performing motor tasks. Stevens et al. [9], Wiley et al. [16] found that obsessive-compulsive disorder, characterized by recurrent and disturbing obsessions and repetitive behaviors, is often a concurrent diagnosis with RSTS and can impair an individual's ability to interact with others and perform the daily life activities. The studies by Verhoeven et al. [17] and Waite et al. [18] confirmed some previously known behavioral features in children with RSTS, such as hyperactivity, emotional dysregulation and self-stimulatory behavior. In addition, their findings introduced a further significant psychological phenotype: anxiety. This suggests that anxiety is a significant and potentially underexplored aspect of the behavioral profile in children with RSTS, further contributing to the ongoing challenges they face. Crawford et al. [19] found that children with RSTS had similar levels of anxiety on the panic attack and agoraphobia subscales and obsessive-compulsive subscales of the Spence Child Anxiety Scale - Parent Version compared to children diagnosed with the anxiety disorder.

Understanding the QOL of children with RTS is essential to developing effective supportive interventions that can improve their well-being and help them lead satisfying lives [20]. QOL is focused on the stage to which children experience quality of life in their family environment, as well as how the family has the potential to achieve its goals in the community and society. RSTS significantly affects the QOL of affected children and their families, as emphasized by much research [9, 21 – 22]. Children with RSTS often demonstrate a range of physical and intellectual disabilities, these delays require constant medical care, specialized educational programs and customized therapeutic interventions, which can be physically, emotionally and financially demanding for families [23-24]. Research on QOL and the impact of disease on family functioning focuses on the main areas of life activities that are important to family functioning and which are usually referred to as dimensions or domains. Taken together, these various domains, such as family interaction, parenting, emotional well-being, physical/material well-being and peer support, represent family life in general [25]. Children with RSTS frequently undergo a wide range of health problems, such as heart defects, breathing problems and feeding difficulties. Regular medical consultations, hospitalizations and surgeries are common, increasing the family's stress and emotional burden. Mori et al. [26] examined family QOL among families with a child with a CDKL5 disorder and found that it was generally lower among those receiving social supports, such as foster care services, suggesting that these families may be more burdened by daily care.

The main aim of the study was to evaluate the health problems of children with RSTS, their QOL and the impact of the disease on family functioning. In addition, we investigate whether the comorbidities, autistic behavior and eating problems affect the children's overall QOL. The final aim was also to determine the impact of frequent hospitalizations and parents' inability to rest on the QOL and functioning of the whole family…”

REVIEWER COMMENTS 3

On page 2, line 54, the sentence “RSTS significantly affects the QoL… as emphasized by much research” needs to be supported by references  

 Thank you for this comment. We have supported by references. Please see incorporated changes.

‘…RSTS significantly affects the QOL of affected children and their families, as emphasized by much research [9, 21 – 22]…”

REVIEWER COMMENTS 4

METHODS

The “Methods” section also needs to be revised and organized. I suggest to create a paragraph titled “The Sample,” where both children and parents are better described.

For example for the children:

  • Age (mean and standard deviation)
  • Gender
  • Genes (EP300 or CREBBP)
  • Intellectual disability (yes or no)
  • Autism spectrum disorder (yes or no)
  • Presence of malformations (yes or no)
  • …..

We currently lack information about these children, including their age range.

Perhaps creating a table would be helpful.

For the parents:

  • Age (for instance, literature on chronic diseases suggests that older parents experience more anxiety and stress than younger parents)
  • Are responses to the questionnaire collected only from mothers or also from fathers?
  • Single-parent families?
  • Educational background and socio-cultural level

 Thank you for this comment. Please see the following rewritten methods section. We also added table, created study sample section and moved children’s characteristics according to reviewer suggestion.

“…

Study sample

The average age of the children involved in the study was 7.9 (SD=4.1). The youngest child was 3 years old, while the oldest was 17 years old. Most children (39%) were in the age range of 5-7 years, 54% of the children studied were boys, while 46% were girls.

The study involved 13 parents, the mothers averaged 40.2 ± 9.5 years (minimum age 23 years and maximum age 65 years) and the fathers 42.1 ± 9.7 years (minimum age 26 years and maximum age 71 years), only one caregiver was single parent.

Clinical characteristics of the prenatal and perinatal period of the study children

The study children were diagnosed with RSTS in the 1st year of life (40%), the 2nd year of life (15%), the 3rd year of life (15%), the 4th year of life (15%) or the 5th year of life (8%). The majority (38%) of the children were born at 36 gestation weeks. Almost half (46%) of the parents first noticed RSTS symptoms right after birth, 38% children were born at 36 weeks. In 62% of children, complications developed during or after delivery. The most common of these were: hypoxia (37.5%), jaundice (37.5%), pneumonia (50%) and respiratory distress syndrome (25%). The defects diagnosed right after birth included: laryngomalacia (18%), cryptorchidism (9%), respiratory failure (9%), multiple organ failure (9%), excessive hair growth (9%), vascular ring (9%), mandibular hypoplasia (9%), high-arched palate (27%), Chiari malformation (9%), esotropia (9%), astigmatism (9%), hearing impairment (9%), craniosynostosis (9%), microcephaly (9%), phimosis (9%), pes calcaneus (9%), syndactyly (9%), bicuspid aortic valve (9%), auricular malformations (9%) and facial dysmorphic features (45%). A total of 54% of patients had to undergo surgery due to the defects. Calculated BMI in five children (38.5%) were underweight (3 girls aged 3, 7 and 9 and 2 boys aged 5 and 4). Two children were overweight (15%) (2 boys aged 4 and 7). One child (8%) was obese (a boy aged 11). A total of 32% of patients had weight deficiency. congenital heart defects were reported in three patients (23%). Congenital heart defects were reported in three patients (23%).  These included: type 2 atrial septal defect, false tendon, and bi-cuspid aortic valve. A total of 23% of caregivers reported that their children suffered from recurrent urinary tract infections. Kidney defects were reported in one child. None of the boys included in the study had hypospadias. Two caregivers reported that their children with obstructive sleep apnea are tired, irritable and hyperactive during the day. In a total of 38% of the children included in the study sleep problems other than obstructive sleep apnea were reported. Four parents (80%) cited sudden and frequent wakeups with difficult to soothe crying as the main sleep problem experienced by their children. Patient’s characteristics are shown in Table 1.

Table 1. Study sample characteristics.

1

2

3

4

5

6

7

8

9

10

11

12

13

Gene

CREBBP

CREBBP

CREBBP

CREBBP

EP300

EP300

EP300

EP300

CREBBP

CREBBP

CREBBP

CREBBP

CREBBP

Diagnosis

18th day

3 years

1 year

2 years

4 years

1 year

2 months

5 years

1 year

2 years

5 years

4 years

2.5 years

Current age

7 years

5 years

11 years

3 years

9 years

7 years

11 years

17 years

4 years

7 years

5 years

4 years

13 years

Child’ weight

25-50th centile

3rd centile

50-75th centile

below the 3rd centile

10th centile

below the 3rd centile

90-97th centile

below the 3rd centile

50-70th centile

50-75th centile

25-50th centile

10-25th centile

below the 3rd centile

Child’s height

3rd-10th centile

3rd-10th centile

25th centile

below the 3rd centile

10-25th centile

below the 3rd centile

3rd-10th centile

below the 3rd centile

10-25th centile

below the 3rd centile

3rd centile

90th centile

below the 3rd centile

BMI

16.9

normal

13.6

underweight

19.8

normal

13.2

underweight

14.6

underweight

13.3

underweight

31.4

obesity

18.4

normal

17.1

overweight

19.6

overweight

17

normal

12.4

underweight

17.8

normal

Child’s head circumference

below the 3rd centile

below the 3rd centile

below the 3rd centile

10th centile

below the 3rd centile

above the 99th centile

-

23rd centile

25-50th centile

7th centile

25-50th centile

25-50th centile

62nd centile

Prenatal course

 NAD

NAD

NAD

abnormal foetal size

NAD

NAD

NAD

NAD

NAD

abnormal amniotic fluid volume, pregnancy at risk of miscarriage

pregnancy at risk of miscarriage

NAD

NAD

HBD

39 weeks

36 weeks

40 weeks

36 weeks

36 weeks

35 weeks

39 weeks

39 weeks

40 weeks

38 weeks

39 weeks

36 weeks

36 weeks

Birth weight (g)

10-50th centile

50th centile

10th centile

10-50th centile

50th centile

50th centile

below the 3rd centile

3rd centile

10th centile

10-50th centile

below the 3rd centile

50th centile

50th centile

Postpartum period

jaundice, pneumonia, feeding problems

hypoxia, respiratory distress syndrome, pneumonia,

feeding problems

hypoxia

feeding problems

jaundice, feeding problems

respiratory distress syndrome, vascular ring, feeding problems

feeding problems, reflux

hypoxia, pneumonia, one lung was not functioning, feeding problems

genital defects, feeding problems

jaundice, feeding problems

feeding problems

second superior vena cava, feeding problems

pneumonia, feeding problems

Motor development

delayed

delayed

delayed

delayed

delayed

delayed

delayed

delayed

delayed

delayed

delayed

delayed

delayed

Speech development

normal

lack of speech development

lack of speech development

-

normal

delayed

delayed

normal

lack of speech development

normal

normal

delayed

delayed

Intellectual disability

moderate

moderate

moderate

moderate

moderate

moderate

significant

moderate

moderate

moderate

mild

mild

moderate

Typical facial dysmorphic features

+

+

+

+

+

+

+

+

+

+

+

+

+

Typical dysmorphic limb features – hands, fingers (including thumbs) and feet

+

+

+

+

+

+

+

+

+

+

+

-

+

Typical behavioural phenotype

+

+

-

+

+

+

+

+

+

+

+

+

+

Hearing

bilateral impairment

bilateral impairment

-

-

-

-

-

-

-

-

-

bilateral hearing impairment

-

Eyesight

astigmatism, myopia

astigmatism, esotropia

hyperopia

strabismus, myopia

-

astigmatism, hyperopia

-

astigmatism, hyperopia

esotropia, astigmatism

hyperopia

-

optic nerve hypoplasia

astigmatism, esotropia, hyperopia

Keloid formation

+

+

-

-

-

-

+

-

-

-

-

+

-

Genital abnormalities

-

+

-

+

-

-

+

-

+

+

-

-

-

Kidney defects

+

-

-

-

-

-

-

-

-

-

-

-

-

Heart defects

+

+

-

-

-

-

-

-

-

-

-

+

-

Short stature

-

-

+

+

-

+

+

-

+

+

+

-

+

Abnormalities of hands, including thumbs

+

+

-

+

-

+

+

+

-

+

+

-

+

Abnormalities of feet, including halluces

+

+

-

-

-

+

+

+

-

+

+

-

-

Occlusal abnormalities / dental anomalies

-

+

+

+

+

+

-

+

+

+

+

+

-

Brain MRI scan

hypoplasia of the P1 segment of the left posterior cerebral artery

NAD

NAD

NAD

small non-specific subcortical hyperintense focus in the posterior right parietal lobe

NAD

NAD

NAD

NAD

NAD

delayed white matter myelination

underdevelopment of optic nerves

and the optic chiasm, underdevelopment of the corpus callosum

hypoplasia of the cerebellar vermis, cysts

NAD; nothing abnormal detected, BMI; body mass index, HBD; gestation age

REVIEWER COMMENTS 5

RESULTS

The entire “Clinical Characteristics” section ( PAG.3 , line137) should be moved to the “Methods” section.

 Thank you for this comment. We have moved.

REVIEWER COMMENTS 6

DISCUSSION

The discussion could benefit from revisions and reorganisations. It should be improved to better explain and discuss the document’s findings. I recommend following the results model, discussing in depth the different questionnaires and correlations, including strengths and weaknesses.

In addition, management and therapeutic recommendations should be provided.

Thank you for your comments. We have reorganized and added therapeutic recommendation. Please see incorporated changes in discussion section.

While studies on the QOL of patients with common diseases are readily available, studies focusing on the QOL of patients with rare diseases are much more difficult to find. This study included a highly diverse group of RSTS patients and their parents, emphasizing the unique challenges and variability in this group. Children differed in age, comorbidities, degree of disability, age at diagnosis and living environment. Similarly, parents varied in age, education, environment, access to specialists, and, most critically, attitudes toward their child's illness and level of acceptability.

The study found that the overall QOL for children with Rubinstein-Taybi Syndrome is markedly low. While these children generally perform best in the emotional domain, they experience the most significant challenges in the physical domain. A study reported by Miller et al. [32] assessed the QOL of children with developmental disorders, showing a growing tendency in which emotional aspects are relatively better, while physical health remains a challenge. In a study carried out by Bidzan et al. [21], the  child studied had high scores on the emotionality subscale, medium scores on the activity subscale, medium scores on the sociability subscale and low scores on the shyness subscale of the EAS Temperament questionnaire. An analysis of the scores reported by the parents on the Satisfaction with Life Scale questionnaire indicated that the parents rated their QOL as average. The children with RSTS included in the present study experienced the highest QOL in emotional functioning. The scores reported for the children by their parents were highest for emotional functioning and lowest for physical functioning. The scores reported by the parents on family questionnaire were highest for cognitive functioning and family relationships and lowest for the worry and daily activities domains. Our study found that the older the child with RSTS, the lower their QOL score. According Von der Lippe et al. [33], age is one of the most important determinants of QOL. As children with RSTS grow older, the cumulative effects of intellectual disability, physical limitations, and social challenges may become more pronounced, leading to a progressive decline in QOL. This age-related trend underscores the need for continuous support and tailored interventions that evolve with the child's age and changing needs. In a study by Agborsangaya et al. [34], associations between multimorbidity and health-related quality of life were assessed. In that study, individuals with at least one chronic condition were found to have significantly lower QOL compared with individuals without chronic conditions. Moreover, individuals with multiple (two or more) chronic conditions had significantly low scores. The study also found that individuals with multimorbidity were twice more likely to be hospitalised compared with the healthy individuals included in the study. In the present study, we hypothesised that the presence of a higher number of medical disorders and more frequent hospital stays in children with RSTS can affect the QOL of the children and their families. The study did not confirm the hypothesis. However, we found that children with eating difficulties (vomiting, choking) have significantly lower QOL.

Children with RSTS have dysmorphic facial, hand and foot features. They also have intellectual and psychomotor delay. In some children speech never develops at all and some children have behavioural problems [35]. Studies have found that approximately 25% of parents of children with RSTS report behavioural problems in their children. These mainly include short attention span, stubbornness, lack of persistence, continuous need for attention, sudden mood changes, resistance to change and repetitive movements [36]. In Grunow et al. [37] study, as many as 65% of children with RSTS were reported to display such unusual behaviours as rocking, spinning and hand flapping. Some children with RSTS exhibit autistic-like behaviours. In the present study, behavioural problems (e.g. irritability, stubbornness, lack of persistence, sudden mood changes) were reported in of the children studied. Autism-like behaviours can increase the already severe symptoms of RSTS, potentially leading to more severe difficulties in social interactions, communication and daily functioning. These behaviours likely contribute to added stress and emotional strain on the family, further compromising their overall QOL. The positive correlation between receiving support from family and friends and higher QOL might be a positive contribution. This finding is consistent with the existing work on chronic disease, which consistently shows that social support has a critical role in alleviating stress and improving overall well-being. For families of children with RSTS, support from extended family, friends and community resources can provide much-needed respite, emotional encouragement and practical help, thereby enhancing their ability to cope with daily challenges. However, the lack of a demonstrated correlation between worsening family relationships and reduced QOL is rather intriguing (p>0.05), in contrast to our expectations. This suggests that while the presence of a child with RSTS clearly affects family dynamics, the direct impact on QOL may be mitigated by other variables, such as dealing strategies, external support and individual family immunity. This finding highlights the need for further research into the specific aspects of family relationships that are most impacted and how they interact with other support systems to influence QOL. Together, these insights underscore the multi-dimensional nature of living with RSTS and the importance of a holistic approach to support. Healthcare providers should be aware of the significant impact of autism-like behaviours and prioritize interventions that address these symptoms. Additionally, qualitative study could reveal the nuanced ways how family dynamics interact with other factors affecting QOL, offering a more comprehensive understanding of the lived experiences of these families. Children with RSTS normally experience intellectual delay, and studies [38] show that the average IQ of patients with RSTS ranges from 35 to 50. In our study, we determined the degree of intellectual disability plays a significant role in the overall QOL of children. Intellectual delay can affect various aspects of daily life, including education, social interaction and independence, thus contributing to the lower QOL observed in our participants. The physical symptoms of RSTS play a role in QOL. In a series of studies of people with RSTS published in 1990 by Rubinstein [1], wide thumbs and/or halluces were reported in 100% of cases. Other commonly found findings included angular deformity of the thumbs with abnormal proximal phalanx shape, angular deformity of the halluces with abnormal shape of the proximal phalanx or first metatarsal bone, clinodactyly of the fifth toe, overlapping toes and broad terminal phalanges in other toes. These physical abnormalities can contribute to functional limitations and social stigmatization, further affecting the QOL for the child and his family.

Since children with RSTS suffer from several congenital anomalies and medical disorders, their caregivers must regularly monitor the health status of their children and have regular contact with many specialists in different medical fields. The medical disorders affecting children with RSTS result in 10 times the average number of hospitals stays as the general population [39]. Considering the poor weight gain often observed in children with RSTS, it is essential to monitoring their growth from birth through regular measurement of weight, height and head circumference. In addition, young patients with RSTS need to be subjected to a variety of specialized tests to meet their specific health needs. Due to the psychomotor delays associated with RSTS, it is often necessary to integrate rehabilitation into the treatment process [40]. 

The present study showed that the families of children exhibiting autistic-like behaviours, such as failure to respond to instructions, lack of peer play and difficulties expressing emotions. The same relationship was found for the child’s QOL. Children with autistic-like behaviours have lower QOL. In a study by Czenczek et al. [41], the QOL of parents of children with autism was examined. In that study, parents of autistic children reported poorer functioning in the areas of general health, mental health, social functioning, energy and vitality. A study by Lee et al. [42] too found that autism has a negative impact on the quality of life of the families of affected children. The study found that the burden of care reported by families with autistic children was significantly higher relative to two comparison groups.

Support from family and friends significantly improves the QOL of both children with RSTS and their parents. Parents who received such support reported higher family impact scores. Marie et al. [32] emphasized that close family members are key to successful treatment and rehabilitation, highlighting the importance of a strong social support network. Similarly, Jamieson et al. [43] found that families with children with disabilities often experience low QOL, particularly in terms of social support. Our results are consistent with these studies, showing that a well-established support can ease stress and improve the overall well-being of families of children with RSTS. As many as 51% of the patients included in the study reported experiencing an increased level of sleepiness during the day, which significantly impaired their functioning. The authors noted that uninterrupted sleep is essential for normal functioning. In the present study, we hypothesised that obstructive sleep apnoea in children with RSTS has a negative impact on the QOL of the children and the functioning of their families.

Research on QOL, especially among all patient groups, not just those with common conditions, is essential. Such studies help identify factors that affect QOL and enable the development of targeted measures to improve it. Understanding the diverse experiences and needs of families affected by RSTS can provide information on better support, health care policies and interventions, ultimately improving their quality of life. Moreover, it should be noted that parents of children with disabilities often feel tired and burdened from constantly caring for their child. Many parents do not receive support from their family and friends and are left on their own, which significantly constrains the time they have to themselves.

Limitation

The study group was not large. However, it should be noted that RSTS is a rare genetic disorder, occurring in about 1 in 100,000 to 125,000 births. Considering the prevalence of RSTS, this represents a significant sample size. Moreover, it is important to consider the serious challenges faced by parents of children with disabilities, including those with RTS. These parents often experience significant fatigue and strain due to the constant care their children require. Many parents do not receive sufficient support from family and friends, causing them to have to handle caregiving responsibilities on their own. This deficiency in support significantly limits their personal time, which makes their participation in our study even more remarkable.

Therapeutic recommendations for children with Rubinstein-Taybi Syndrome

Systematic reassessment of the child's development and the effectiveness of interventions is necessary, allowing the therapeutic plan to be adjusted as the child grows and his needs change. Therapy should be initiated as early as possible to support cognitive, motor, and social development. Early intervention is crucial for improving long-term outcomes and maximizing the child’s potential. A multidisciplinary care approach is essential, involving a team of specialists such as pediatricians, geneticists, neurologists, speech therapists, occupational therapists, nursing care and physical therapists. Physical therapy focuses on strengthening muscles, improving motor skills, and enhancing overall mobility. Occupational therapy aims to enhance fine motor skills, daily living activities, and sensory processing. It involves targeted activities to improve hand-eye coordination, feeding abilities, and self-care skills. Sensory integration therapy can also be effective in managing sensory processing challenges. Speech and language therapy is essential for advancing both verbal and non-verbal communication skills and addressing issues related to feeding and swallowing.

REVIEWER COMMENTS 7

ABSTRACT

The abstract will have to be rewritten after substantial changes to the structure of the article.

Abstract: Background/Objectives: Rubinstein-Taybi Syndrome (RSTS- OMIM, #180849) is a rare genetic disorder associated with distinctive clinical features, including a typical craniofacial appearance, global developmental delay, intellectual disability and broad, angular thumbs and fingers. The main aim of the study was to evaluate the health problems of children with RTST, their quality of life and the impact of the disease on family functioning. In addition, we investigate whether the comorbidities, autistic behavior and eating problems affect the children's overall QOL. Methods: A cross-sectional study was performed, including a total of 13 caregivers of children diagnosed with RSTS. A self-reported questionnaire [SRQ], medical records and the Pediatric Impact Module PedsQLTM 2.0, the Pediatric Quality of Life PedsQLTM 4.0 were used to obtain data on QOL and the impact of the disease on family functioning. Results: The overall QOL score for children with RSTS was x=52.40; SD 13.01. The highest QOL was in emotional functioning (EF; x=59.23; SD 18.69), while the lowest QOL was in physical functioning (PF; x=48.56; SD 16.32) and social functioning (SF; x=48.85; SD 18.84). There was a statistically significant negative correlation (p<0.03; r=-2.01) between the age of the child and their QOL, indicating that older children had lower QOL scores. The mean overall rating for the impact of RSTS on family functioning was x=50.00; SD 10.91. Caregivers reported the highest scores for cognitive functioning (CF; x=64.23; SD 23.70) and family relationships (FR; x=60.00; SD 17.17). The lowest scores were for daily activities (DA; x=41.03; SD 17.17) and worry (W; x=37.69; SD 18.55). Conclusions: This study provides the first comprehensive exploration of the QOL of children with RSTS) and its impact on family functioning.

REVIEWER COMMENTS 8

ADDITIONAL ELEMENTS

  • Examine the bibliography carefully.

For example: several references need to be corrected, such as the numbers 24 and 25.

Thank you for your comments. We have corrected.

Reviewer 2 Report

Comments and Suggestions for Authors

This article presented by Rozensztrauch et al. proposes a study entitled “Rubinstein-Taybi syndrome clinical characteristics from the perspective of quality of life and the impact of the disease on family functioning” which aims to evaluate the quality of life of 14 children and the impact of the disease on families to improve their well-being through customized care and family support.

Despite the authors' intent to address the important issue of the quality of life of patients with Rubinstein-Taybi syndrome and their caregivers, the work exhibits inconsistency.  Although the authors have compiled detailed information on the patients and 13 caregivers, the sample size is limited and remains inadequate.

A number of points have been discussed, but they do not offer any obvious novelty or results pertaining to the situation.

Please take note that in OMIM, a gene is denoted by an asterisk (*) before the entry number, whereas a descriptive entry, which typically pertains to a phenotype and not a unique locus, is denoted by a number symbol (#) before the entry number. So, please add the #.

The abbreviations RSTS and RST are used interchangeably without consistency.

Kindly ensure that the various percentages in the text are verified for accuracy. 

Additionally, please note that the titles of the tables are inconsistently presented, as some are displayed as "Table 3." while others are denoted as ".Table nr 5".

Also, repetitive sentences are present, such as "The majority (38%) of the children were born at 36 gestation weeks" (line 141) and "The largest proportion of the 142 children (38%) were born at 36 weeks" (line 143).

Part of the text includes instructions from the template (lines 217-219).

Comments on the Quality of English Language

I would like to suggest a comprehensive revision of the approach to composing English language.

Author Response

Dear Reviewer 2,

Thank you very much for sending us the consensus opinion about requested revision of our manuscript entitled: Rubinstein-Taybi syndrome clinical characteristics from the perspective of quality of life and the impact of the disease on family functioning. We appreciate the thoughtful comments, and we have modified the manuscript in response to your suggestions, which we believe will further improve its quality. 

REVIEWER COMMENTS 1

This article presented by Rozensztrauch et al. proposes a study entitled “Rubinstein-Taybi syndrome clinical characteristics from the perspective of quality of life and the impact of the disease on family functioning” which aims to evaluate the quality of life of 14 children and the impact of the disease on families to improve their well-being through customized care and family support. Despite the authors' intent to address the important issue of the quality of life of patients with Rubinstein-Taybi syndrome and their caregivers, the work exhibits inconsistency.  Although the authors have compiled detailed information on the patients and 13 caregivers, the sample size is limited and remains inadequate. A number of points have been discussed, but they do not offer any obvious novelty or results pertaining to the situation.

Thank you for your comments. Please see incorporated changes to improve the quality of the manuscript.

REVIEWER COMMENTS 2

Please take note that in OMIM, a gene is denoted by an asterisk (*) before the entry number, whereas a descriptive entry, which typically pertains to a phenotype and not a unique locus, is denoted by a number symbol (#) before the entry number. So, please add the #.

Thank you for your comments. We have corrected. Please see changes.

..”Rubinstein-Taybi Syndrome (RSTS- OMIM, #180849) is a rare genetic disorder, named after the doctors who first described it, Dr. Jack Rubinstein and Dr. Hooshang Taybi, in 1963 [1]…”

REVIEWER COMMENTS 3

The abbreviations RSTS and RST are used interchangeably without consistency.

Thank you for your comments. We have corrected through manuscript.

REVIEWER COMMENTS 4

Kindly ensure that the various percentages in the text are verified for accuracy. Additionally, please note that the titles of the tables are inconsistently presented, as some are displayed as "Table 3." while others are denoted as "Table nr 5".

Thank you for your comments. We have corrected

REVIEWER COMMENTS 5

Also, repetitive sentences are present, such as "The majority (38%) of the children were born at 36 gestation weeks" (line 141) and "The largest proportion of the 142 children (38%) were born at 36 weeks" (line 143).

Thank you for your comments, we have rewritten whole section an incorporated change.

”Study sample

The average age of the children involved in the study was 7.9 (SD=4.1). The youngest child was 3 years old, while the oldest was 17 years old. Most children (39%) were in the age range of 5-7 years, 54% of the children studied were boys, while 46% were girls.

The study involved 13 parents, the mothers averaged 40.2 ± 9.5 years (minimum age 23 years and maximum age 65 years) and the fathers 42.1 ± 9.7 years (minimum age 26 years and maximum age 71 years), only one caregiver was single parent.

Clinical characteristics of the prenatal and perinatal period of the study children

The study children were diagnosed with RSTS in the 1st year of life (40%), the 2nd year of life (15%), the 3rd year of life (15%), the 4th year of life (15%) or the 5th year of life (8%). The majority (38%) of the children were born at 36 gestation weeks. Almost half (46%) of the parents first noticed RSTS symptoms right after birth, 38% children were born at 36 weeks. In 62% of children, complications developed during or after delivery. The most common of these were: hypoxia (37.5%), jaundice (37.5%), pneumonia (50%) and respiratory distress syndrome (25%). The defects diagnosed right after birth included: laryngomalacia (18%), cryptorchidism (9%), respiratory failure (9%), multiple organ failure (9%), excessive hair growth (9%), vascular ring (9%), mandibular hypoplasia (9%), high-arched palate (27%), Chiari malformation (9%), esotropia (9%), astigmatism (9%), hearing impairment (9%), craniosynostosis (9%), microcephaly (9%), phimosis (9%), pes calcaneus (9%), syndactyly (9%), bicuspid aortic valve (9%), auricular malformations (9%) and facial dysmorphic features (45%). A total of 54% of patients had to undergo surgery due to the defects. Calculated BMI in five children (38.5%) were underweight (3 girls aged 3, 7 and 9 and 2 boys aged 5 and 4). Two children were overweight (15%) (2 boys aged 4 and 7). One child (8%) was obese (a boy aged 11). A total of 32% of patients had weight deficiency. congenital heart defects were reported in three patients (23%). These included: type 2 atrial septal defect, false tendon, and bicuspid aortic valve. A total of 23% of caregivers reported that their children suffered from recurrent urinary tract infections. Kidney defects were reported in one child. None of the boys included in the study had hypospadias. Two caregivers reported that their children with obstructive sleep apnoea are tired, irritable and hyperactive during the day. In a total of 38% of the children included in the study sleep problems other than obstructive sleep apnoea were reported. Four parents cited sudden and frequent wakeups with difficult-to-soothe crying as the main sleep problem experienced by their children. Patient’s characteristics are shown in Table 1….”

REVIEWER COMMENTS 6

Part of the text includes instructions from the template (lines 217-219).

Thank you very much for these comments. We have deleted from manuscript.

Round 2

Reviewer 2 Report

Comments and Suggestions for Authors

The article by Rozensztrauch et al. proposes a study entitled “Rubinstein-Taybi syndrome clinical characteristics from the perspective of quality of life and the impact of the disease on family functioning” which aims to appraise the well-being of RSTS children and the impact of the condition on family functioning in order to enhance their quality of life through customized care and family support.

The authors stress the importance of consistently showing social support to RSTS families, which is crucial for reducing stress and enhancing overall well-being. Despite the authors' comprehensive compilation of detailed information on 13 patients and their caregivers, it is important to note that the sample size remains limited. Nevertheless, this study serves as a commendable initial exploration of the topic of quality of life in RSTS, warranting consideration for future in-depth investigation.

The work has been significantly refined and improved in terms of its structure and the use of the english language. 

Major comments:

  • Unfortunately, Table 1 is not included in the text.

Minor comments:

  • Check the nomenclature to correctly write gene names (e.g., line 41 "CREBBP", gene symbols are italicized).
  • The authors describe 13 caregivers, and note that only one of them is a single parent. It would have been valuable to have the opportunity to evaluate the perspectives of both parents wherever feasible.
  • Repetitive sentences are present: Line 124 "38% children were born at 36 weeks" is a repetition of "The majority (38%) of the children 122 were born at 36 gestation weeks." (line 122-123)
  • Line 134 "congenital" should have a capital letter
  • Line 328: please correct RSTS instead of RST to be consistent with the text.

Author Response

Dear Reviewer 1,

Thank you very much for sending us the consensus opinion about requested revision of our manuscript entitled: Rubinstein-Taybi syndrome clinical characteristics from the perspective of quality of life and the impact of the disease on family functioning. We appreciate the thoughtful comments, and we have modified the manuscript in response to your suggestions, which we believe will further improve its quality. 

REVIEWER COMMENTS 1

The article by Rozensztrauch et al. proposes a study entitled “Rubinstein-Taybi syndrome clinical characteristics from the perspective of quality of life and the impact of the disease on family functioning” which aims to appraise the well-being of RSTS children and the impact of the condition on family functioning in order to enhance their quality of life through customized care and family support. The authors stress the importance of consistently showing social support to RSTS families, which is crucial for reducing stress and enhancing overall well-being. Despite the authors' comprehensive compilation of detailed information on 13 patients and their caregivers, it is important to note that the sample size remains limited. Nevertheless, this study serves as a commendable initial exploration of the topic of quality of life in RSTS, warranting consideration for future in-depth investigation.

The work has been significantly refined and improved in terms of its structure and the use of the english language. 

Thank you for your comments

Major comments:

Unfortunately, Table 1 is not included in the text.

Thank you for your comments, Of course, the table will be an integral part of the manuscript, only that, due to the size of the tamplate, it will be placed by the editor in the right place.

Minor comments:

Check the nomenclature to correctly write gene names (e.g., line 41 "CREBBP", gene symbols are italicized).

Thank you for your comments. We have corrected

The authors describe 13 caregivers and note that only one of them is a single parent. It would have been valuable to have the opportunity to evaluate the perspectives of both parents wherever feasible.

Thank you very much for the right comment, of course we will try to ask both parents in the future, but the other parent could not be approached.

Repetitive sentences are present: Line 124 "38% children were born at 36 weeks" is a repetition of "The majority (38%) of the children 122 were born at 36 gestation weeks." (line 122-123),

Thank you for your comments. We have corrected

Line 134 "congenital" should have a capital letter

Thank you for your comments. We have corrected

Line 328: please correct RSTS instead of RST to be consistent with the text.

Thank you for your comments. We have corrected
